# An adaptable implementation package targeting evidence-based indicators in primary care: A pragmatic cluster-randomised evaluation

**Thomas A. Willis**[1]\*, **Michelle Collinson**[2], **Liz Glidewell**[3], **Amanda J. Farrin**[2], **Michael Holland**[2], **David Meads**[1], **Claire Hulme**[4], **Duncan Petty**[5], **Sarah Alderson**[1], **Suzanne Hartley**[2], **Armando Vargas-Palacios**[1], **Paul Carder**[6], **Stella Johnson**[6], **Robbie Foy**[1], **on behalf of the ASPIRE programme team**¶

1 Leeds Institute of Health Sciences, University of Leeds, Leeds, United Kingdom, 2 Clinical Trials Research Unit, Leeds Institute of Clinical Trials Research, University of Leeds, Leeds, United Kingdom, 3 Department of Health Sciences, Hull York Medical School, University of York, York, United Kingdom, 4 College of Medicine and Health, University of Exeter, Exeter, United Kingdom, 5 School of Pharmacy and Medical Sciences, University of Bradford, Bradford, United Kingdom, 6 West Yorkshire Research and Development, NHS Bradford Districts CCG, Bradford, United Kingdom

¶ Membership of the ASPIRE programme team is provided in the Acknowledgements.
\* t.a.willis@leeds.ac.uk

**Data Availability Statement:** Data cannot be shared publicly owing to a need to maintain patient confidentiality. Interested researchers may contact

## Abstract

### Background

In primary care, multiple priorities and system pressures make closing the gap between evidence and practice challenging. Most implementation studies focus on single conditions, limiting generalisability. We compared an adaptable implementation package against an implementation control and assessed effects on adherence to four different evidence-based quality indicators.

### Methods and findings

We undertook two parallel, pragmatic cluster-randomised trials using balanced incomplete block designs in general practices in West Yorkshire, England. We used 'opt-out' recruitment, and we randomly assigned practices that did not opt out to an implementation package targeting either diabetes control or risky prescribing (Trial 1); or blood pressure (BP) control or anticoagulation in atrial fibrillation (AF) (Trial 2). Within trials, each arm acted as the implementation control comparison for the other targeted indicator. For example, practices assigned to the diabetes control package acted as the comparison for practices assigned to the risky prescribing package. The implementation package embedded behaviour change techniques within audit and feedback, educational outreach, and computerised support, with content tailored to each indicator. Respective patient-level primary endpoints at 11 months comprised the following: achievement of all recommended levels of haemoglobin A1c (HbA1c), BP, and cholesterol; risky prescribing levels; achievement of recommended BP; and anticoagulation prescribing. Between February and March 2015, we

ctru-dataaccess@leeds.ac.uk to request and obtain relevant data.

**Funding:** This study was funded by the National Institute for Health Research (NIHR) (Programme Grants for Applied Research [Grant Reference Number RP-PG-1209-10040], Principal Investigator = RF), https://www.nihr.ac.uk/. The funders had no role in study design, data collection and analysis, decision to publish, or preparation of the manuscript.

**Competing interests:** I have read the journal's policy and the authors of this manuscript have the following competing interests: PC and SJ are employed by NHS Bradford Districts CCG, which commissioned and funded Prescribing Support Services Ltd. (PSS; DP's practice pharmacy company) to conduct the educational outreach visits as part of the implementation package. The commission followed established NHS commissioning guidelines. DP was responsible for the recruitment and training of pharmacist-facilitators to deliver the educational outreach visits. Neither he nor the facilitators were involved in the analysis. PC and SJ were not involved in the implementation or analysis of the study. SA was funded by an NIHR academic clinical lectureship (2014-2019). RF received grant funding from NIHR Programme Grants for Applied Research during the conduct of this study. There are no other relationships or activities that could appear to have influenced the submitted work. All authors have completed the ICMJE uniform disclosure form at www.icjme.org/coi_disclosure.pdf.

**Abbreviations:** ACE-I, angiotensin-converting-enzyme inhibitor; AF, atrial fibrillation; ARB, angiotensin receptor blocker; BP, blood pressure; CCG, clinical commissioning group; $CHA_2DS_2$-VASc, congestive heart failure, hypertension, age $\geq$75, diabetes, stroke, vascular disease, age between 65 and 74, and female sex; CI, confidence interval; CKD, chronic kidney disease; GP, general practitioner; HbA1c, haemoglobin A1c; ICC, intra-cluster correlation coefficient; IMD, Index of Multiple Deprivation; ITT, intention-to-treat; NHS, National Health Service; NICE, National Institute for Health and Care Excellence; NSAID, nonsteroidal anti-inflammatory drug; OR, odds ratio; QALY, quality-adjusted life year; QOF, Quality Outcomes Framework; UKPDS-OM, UK Prospective Diabetes Study Outcomes Model.

recruited 144 general practices collectively serving over 1 million patients. We stratified computer-generated randomisation by area, list size, and pre-intervention outcome achievement. In April 2015, we randomised 80 practices to Trial 1 (40 per arm) and 64 to Trial 2 (32 per arm). Practices and trial personnel were not blind to allocation. Two practices were lost to follow-up but provided some outcome data. We analysed the intention-to-treat (ITT) population, adjusted for potential confounders at patient level (sex, age) and practice level (list size, locality, pre-intervention achievement against primary outcomes, total quality scores, and levels of patient co-morbidity), and analysed cost-effectiveness. The implementation package reduced risky prescribing (odds ratio [OR] 0.82; 97.5% confidence interval [CI] 0.67–0.99, $p = 0.017$) with an incremental cost-effectiveness ratio of £1,359 per quality-adjusted life year (QALY), but there was insufficient evidence of effect on other primary endpoints (diabetes control OR 1.03, 97.5% CI 0.89–1.18, $p = 0.693$; BP control OR 1.05, 97.5% CI 0.96–1.16, $p = 0.215$; anticoagulation prescribing OR 0.90, 97.5% CI 0.75–1.09, $p = 0.214$). No statistically significant effects were observed in any secondary outcome except for reduced co-prescription of aspirin and clopidogrel without gastro-protection in patients aged 65 and over (adjusted OR 0.62; 97.5% CI 0.39–0.99; $p = 0.021$). Main study limitations concern our inability to make any inferences about the relative effects of individual intervention components, given the multifaceted nature of the implementation package, and that the composite endpoint for diabetes control may have been too challenging to achieve.

## Conclusions

In this study, we observed that a multifaceted implementation package was clinically and cost-effective for targeting prescribing behaviours within the control of clinicians but not for more complex behaviours that also required patient engagement.

## Trial registration

The study is registered with the ISRCTN registry (ISRCTN91989345).

## Author summary

### Why was this study done?

- Commonly used interventions to implement evidence-based practice, e.g., audit and feedback, educational outreach, and computerised prompts, generally have modest if variable effects on clinical performance.

- The effects of such interventions may be enhanced by tailoring them to identified needs and barriers.

- Trials of implementation interventions typically address single conditions; it is difficult to judge whether an intervention that works for one condition will work for another.

**What did the researchers do and find?**

- We conducted two parallel, pragmatic trials to evaluate an implementation package for primary care that was adapted to overcome barriers for different clinical priorities.

- General practices were randomly assigned to receive an implementation package targeting diabetes control or risky prescribing (Trial 1); blood pressure control or anticoagulation in atrial fibrillation (Trial 2). Respective primary endpoints assessed were as follows: achievement of all recommended levels of haemoglobin A1c, BP, and cholesterol; risky prescribing levels; achievement of recommended BP; and anticoagulation prescribing.

- The implementation package produced a significant clinically and cost-effective reduction in one target only: risky prescribing.

**What do these findings mean?**

- In this study, we found that an adaptable implementation package was cost-effective for targeting prescribing behaviours within the control of clinicians, but not for more complex behaviours that also required patient engagement.

- Given known associations between risky prescribing combinations and increased morbidity, mortality, and health service use, a scaled-up risky prescribing implementation package could have an important population impact.

## Introduction

Clinical research can only benefit patient and population health if findings are incorporated into routine care. There are delays and inappropriate variations in uptake of effective treatments and withdrawal of less effective or harmful treatments [1]. This translation gap is important to policy makers, healthcare systems, and research funders because it limits the health, social, and economic impacts of clinical research [2].

United Kingdom primary care presents particular implementation challenges: growing demand, increasing complexity of care, and limited workforce capacity, against a background of frequent organisational reconfigurations [3,4]. We identified 107 clinical guidelines produced by the National Institute for Health and Care Excellence (NICE) relevant to UK general practice [5]. Many implementation studies focus on a single clinical condition or problem (e.g., diabetes, antibiotic stewardship), limiting generalisability, as it is uncertain whether an implementation strategy developed for one condition will work for another. It is impracticable and inefficient to devise an implementation strategy for every new guideline. Furthermore, the clinical significance of behaviours often targeted in implementation studies, such as receipt of processes of care or investigations, is doubtful [6]. Implementation strategies are required, capable of integration into primary care systems and adaptable to different clinical priorities.

We earlier derived evidence-based indicators that could be measured using routinely collected data [5]. We found marked variations in indicator achievement and scope for improvement in a sample of 89 general practices [7]. We subsequently focused our efforts on four 'high

impact' indicators that would benefit population health if implemented more consistently: achievement of recommended treatment targets for all of haemoglobin A1c (HbA1c), blood pressure (BP), and cholesterol in type 2 diabetes [8]; avoidance of risky prescribing of nonsteroidal anti-inflammatory drugs (NSAIDs) and anti-platelet drugs [9]; achievement of recommended BP levels in patients at high risk of cardiovascular events [10]; and anticoagulant prescribing for stroke prevention in atrial fibrillation (AF) [11]. We drew upon systematic reviews of implementation strategies [6,12–14], theory-guided interviews with primary care staff [15], and workshops with health professionals and patients to develop a multifaceted implementation package [16]. We compared the effects of the adaptable implementation package against implementation control on adherence to four different evidence-based high impact indicators.

## Methods

### Study design and participants

We conducted two parallel, cluster-randomised trials using balanced incomplete block designs. In implementation trials, there may be positive attention or negative demotivation effects from participant knowledge of allocation to an intervention or control group, respectively. Balanced incomplete block designs aim to balance any such nonspecific effects across trial arms, as each arm receives an intervention, thereby increasing confidence that any difference in outcomes is attributable to the intervention [17]. The design is incomplete, as each arm receives only one of the interventions. General practices providing National Health Service (NHS) care were the unit of allocation, as the intervention was delivered at the practice level. We maximised pragmatism in trial design and execution to ensure 'real-world' relevance [18].

Practices were recruited from West Yorkshire, England, which covers a diverse population of 2.2 million residents, albeit with deprivation levels above the national average [19]. Around 300 general practices were then organised within 10 clinical commissioning groups (CCGs).

Eligible general practices used *SystmOne*, the computerised clinical system used by approximately two thirds of West Yorkshire practices (The Phoenix Partnership, http://www.tpp-uk. com). We excluded 31 practices that had contributed to earlier intervention development work. We invited practices via recorded post and email, with reminders at two weeks. The UK National Research Ethics Service granted ethical approval (14/SC/1393). This permitted use of an opt-out approach: all eligible practices were included, unless actively declined within four weeks of invitation. The study protocol has previously been published [20]. The study is registered with the ISRCTN registry (ISRCTN91989345). This study is reported as per the CONSORT guideline for cluster-randomised trials (S1 CONSORT checklist).

### Randomisation and masking

The trial statistician performed the two-stage randomisation using a bespoke computer-generated minimisation programme implemented in R, software version 2.15.2 (R Foundation for Statistical Computing, Vienna, Austria) (incorporating a random element). First, practices were stratified by CCG and list size, and randomised to Trial 1 (implementation packages targeting either diabetes control or risky prescribing), Trial 2 (BP control or anticoagulation in AF), or no intervention (80:64:34). As recruitment exceeded expectations, we included a 'no intervention' arm to allow us to evaluate any nonspecific effects of research participation (this paper presents comparisons within Trials 1 and 2; analysis of the no intervention group will be reported elsewhere). Secondly, within each trial we stratified practices by CCG, list size, and pre-intervention adherence to the two relevant targeted indicators and randomised them 1:1 to individual trial arms (Fig 1). Within each trial, we assumed that any

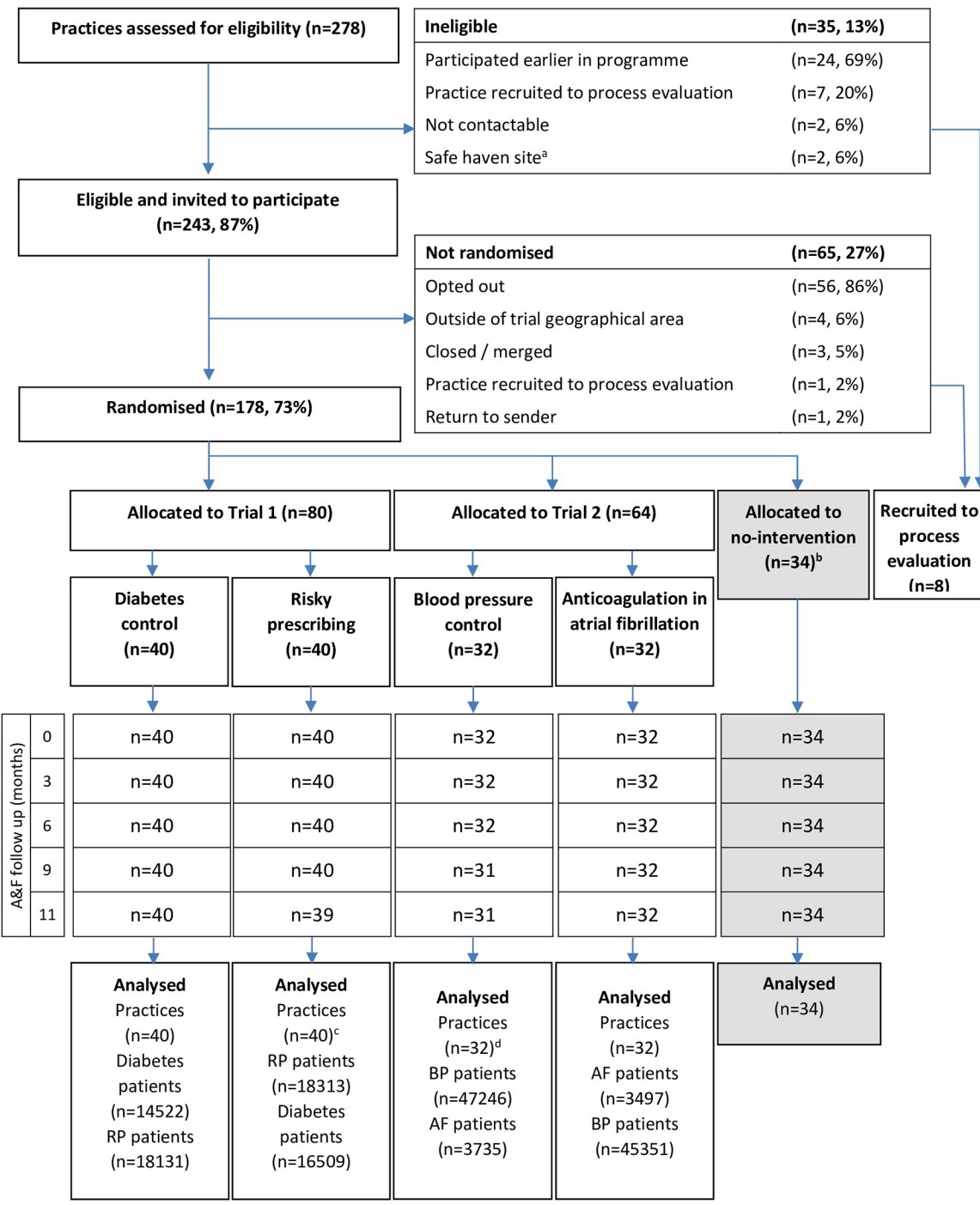

**Fig 1. Trial profile.** [a]Safe haven practices are for those patients who have demonstrated violent tendencies, who have been removed from usual general practice, or for ex-offenders. [b]The flow diagram presents the whole randomisation process, but this paper reports the planned comparisons within Trials 1 and 2. The no intervention arm is shaded to indicate that it was not included in the analyses described here; analysis of this group will be reported elsewhere. [c]One practice in the risky prescribing arm merged with a non-ASPIRE practice in advance of the fourth feedback report. However, as they received the first three feedback reports, some outcome data are available, and they are included in the final analyses. [d]One practice in the BP control arm closed in advance of the third feedback report. However, as they received the first two feedback reports, some outcome data are available and they are included in the final analyses. As an example, for the diabetes control practices: there were 14,522 patients acting as the intervention group for the diabetes control implementation package and 18,131 patients acting as the control group for the risky prescribing implementation package. A&F follow-up: delivery of audit and feedback reports during the intervention period; AF, atrial fibrillation; BP, blood pressure; RP, risky prescribing.

clinical effects of either implementation package would be independent of one another; each intervention arm therefore acted as the control arm for the other intervention arm in the same trial. For example, practices assigned to the diabetes control package acted as the implementation control comparison for practices assigned to the risky prescribing package. Practices from one CCG (29 practices) involved in a concurrent initiative targeting one of our study priorities, anticoagulation in AF, were excluded from Trial 2. Practices and trial personnel were not blind to allocation.

General practice and trial personnel involved in intervention delivery and trial statisticians responsible for analysis were, of necessity, aware of allocation assignment. Data collection for all endpoints was masked to allocation.

## Procedures

The implementation package (detailed elsewhere [16]) embedded behaviour change techniques within typical primary care interventions: audit and feedback, educational outreach, and computerised support. Implementation package content, specifically the clinical content and embedded behaviour change techniques, was adapted for each of the four targeted indicators and delivered to practices May 2015 to March 2016.

Audit and feedback aimed to give comparative feedback on achievement, inform and prompt recall of clinical goals, highlight consequences of changing or not changing practice, suggest strategies for change, and encourage goal setting and reflection on progress towards goals. Quarterly electronic and paper practice–specific reports presented achievement ranked by practice and compared over time for relevant trial indicators in graphical and numerical forms, using remotely gathered, individualised practice data. Reports also contained brief, evidence-based clinical messages; responses to common queries; and action-planning templates. Practices received computerised search tools to identify relevant patients for review. We encouraged practices to use the feedback reports as supporting materials for professional appraisal and revalidation.

Educational outreach aimed to enhance feedback reports by facilitating individual and group reflection, discussing barriers to action, sharing models of good practice, enhancing motivation, and action planning. We trained pharmacists over two days to deliver 30-minute sessions, designed to fit with existing practice meetings and offered to all staff involved in patient and practice management. We identified a key clinical contact to support practice engagement. We offered a second, follow-up session to review progress and refine action plans, as well as two days of pharmacist support for patient identification and review.

Prompts and reminders reinforced clinical messages and indicator adherence. Computerised prompts for risky prescribing were offered to practices and had to be accepted before they became active on their system. If accepted, they were triggered during consultations and repeat prescribing through an algorithm for patient age, diagnosis, drug, and duration. A one-click justification (ignore, or add or stop medication) was required before users could proceed. Other targeted indicators had existing prompts on practice computer systems, usually to support the Quality Outcomes Framework (QOF), a performance management system whereby general practices are remunerated according to achievement of targets reflecting quality of care [21]. We provided laminated reminders to convey key clinical information (e.g., management pathways) for BP control, anticoagulation for AF, and risky prescribing. We also developed checklists to facilitate shared decision-making with patients for managing BP and diabetes, but we could not make them available in a satisfactory format within the study time lines [16].

We deliberately aligned intervention delivery with the QOF year to fit with existing practice schedules. We assumed that any actions prompted by the implementation package would occur within this period.

## Outcomes

Anonymised patient-level outcome data were obtained remotely from general practices from *SystmOne*. We planned to measure all outcomes at 12 months post-randomisation. However, a one-month delay in randomisation effectively meant that we were only able to assess outcomes at 11 rather than 12 months.

The primary outcomes for each intervention arm comprised the following:

**Trial 1.** • Proportion of patients with type 2 diabetes achieving all three treatment targets:

   a. BP below 140/80 mmHg (or 130/80 mmHg if kidney, eye, or cerebrovascular damage);

   b. HbA1c value below or equal to 59 mmol/mol;

   c. cholesterol level below or equal to 5.0 mmol/L.

• Proportion of patients meeting at least one of nine indicators of high-risk NSAID and anti-platelet prescribing:

   a. prescription of a traditional oral NSAID or low-dose aspirin in patients with a history of peptic ulceration without co-prescription of gastro-protection;

   b. traditional oral NSAID in patients aged 75 years or over without co-prescription of gastro-protection;

   c. traditional oral NSAID and aspirin in patients aged 65 years or over without co-prescription of gastro-protection;

   d. aspirin and clopidogrel in patients aged 65 years or over without co-prescription of gastro-protection;

   e. warfarin and traditional oral NSAID;

   f. warfarin and low-dose aspirin or clopidogrel without co-prescription of gastro-protection;

   g. oral NSAID in patients with heart failure;

   h. oral NSAID in patients prescribed both a diuretic and an angiotensin-converting-enzyme inhibitor (ACE-I) or angiotensin receptor blocker (ARB);

   i. oral NSAID in patients with chronic kidney disease (CKD).

**Trial 2.** • Proportion of patients achieving the lowest appropriate BP target:

   a. under 140/90 mmHg if aged under 80 years with hypertension, coronary heart disease, peripheral arterial disease, a history of stroke or transient ischemic attack, or a 10-year cardiovascular disease risk of 20% or higher;

   b. under 150/90 mmHg if aged 80 years and over with hypertension;

   c. under 140/80 mmHg if aged under 80 years with diabetes, under 130/80 mmHg if complications of diabetes or aged under 80 years with CKD and proteinuria.

- Combined proportion of men with AF and a congestive heart failure, hypertension, age ≥75, diabetes, stroke, vascular disease, age between 65 and 74, and female sex ($CHA_2DS_2$-VASc) score of 1 and women with a $CHA_2DS_2$-VASc score of 2 or above prescribed anticoagulation therapy.

Secondary outcomes (S1–S4 Tables) included individual indicators contributing to composite primary outcomes, processes of care, and continuous intermediate clinical outcomes (last recorded BP, HbA1c, cholesterol). We assessed QOF indicators relevant to implementation packages (diabetes control, BP control, and anticoagulation for AF) to examine any effects on existing routine indicators. We assessed QOF indicators not directly targeted by the implementation packages to assess any unintended wider impacts. These included coronary heart disease and asthma as example long-term physical conditions, mental health (long-term non-physical condition), and smoking (health promotion).

We obtained practice characteristic data from publicly available sources (Health Education England, Health and Social Care Information Centre): practice list size (number of registered patients); number of general practitioner (GP) partners and salaried GPs; practice training status; practice level Index of Multiple Deprivation (IMD); ethnic profile of practice register; achievement of QOF indicators (2014–2015); patient satisfaction (proportion who would recommend the practice to others); patient-rated practice accessibility (proportion able to speak with GP or nurse within 48 hours of approach); and practice prescribing costs. Patient characteristics (age, sex, comorbidity [number of QOF disease registers on which the patient appeared]) and outcome data were extracted from *SystmOne* by the local primary care commissioning support unit.

We monitored intervention delivery and fidelity via standardised report forms completed by outreach facilitators following each visit. We recorded reasons for declining visits. We monitored which practices accepted the invitation to join indicator-specific *SystmOne* organisational groups to access patient searches and any related computerised prompts.

## Sample size

Median effect sizes on processes and outcomes of care for a range of guideline implementation studies are around 4%–9% [6]. Given our enhanced interventions and targeting of indicators with scope for improvement [12], we judged an absolute difference of 15% for outcomes related to diabetes, BP control, and anticoagulation for AF as realistic and clinically relevant. Baseline achievement rates in risky prescribing were considerably higher and, considering a potential ceiling effect, a 5% decrease was realistic and clinically relevant.

We aimed to recruit 144 practices: 80 in Trial 1 (diabetes and risky prescribing) and 64 in Trial 2 (BP control and AF), giving 90% power at 2.5% significance (adjusting for two comparisons per trial) to detect the specified effect sizes, allowing for 10% attrition. The mean cluster size (number of patients per practice by indicator), coefficient of variation, intra-cluster correlation coefficient (ICC), and control group achievement rates were estimated from previously collected data [7,20] (S5 Table).

## Statistical analysis

Analyses, conducted in SAS software version 9.4 (SAS Institute Inc., Cary, NC) according to the prespecified statistical analysis plan (S1 statistical analysis plan), were based on intention-to-treat (ITT), with two-sided significance testing at the 2.5% level to preserve an overall 5% Type I error rate per trial. We analysed primary outcomes using two-level binary logistic regression models, with patients (level 1) nested within randomised practices (level 2). We

analysed binary secondary outcomes for individual indicators within the composite primary outcomes, and recorded processes of care using similar multilevel logistic models. We analysed continuous intermediate clinical outcomes using two-level linear models. All analyses adjusted for patient-level (sex, age) and practice-level covariates (baseline practice list size, CCG, pre-intervention achievement against primary outcomes, total QOF score 2014–2015 [covering clinical and management domains], proportion of patients with 0–3 comorbidities). Sensitivity analyses were conducted when model assumptions were violated. Data completeness for these analyses depended on the completeness of *SystmOne* medical records and could not be assessed within the trial data set. Missing data in a patient's record could have led to their incorrect inclusion or exclusion from the denominator of a particular quality indicator or resulted in the patient being incorrectly classified as achieving that indicator. For the primary and secondary analyses, we assumed that data were missing at random. Imputation was not performed for any outcome variables. Complete case analyses were performed for the continuous clinical outcomes. No adjustment for multiple comparisons was made in the secondary endpoint analyses.

## Cost-effectiveness

We conducted economic evaluations of three implementation packages using a decision-analytic modelling approach. Given resource constraints, we were unable to evaluate the anticoagulation for AF package. Implementation package costs included researcher and healthcare professional time and materials in delivery and resulting from change in practice. Trial estimates of effectiveness were inputted into decision models, with the main output being cost (UK £, 2017 prices) per incremental quality-adjusted life year (QALY) over a lifetime horizon from the perspective of the UK health and social care provider. Where possible we used models, parameters, and assumptions that had informed the original NICE guidance. For diabetes control, we used the UK Prospective Diabetes Study Outcomes Model (UKPDS-OM) version 2 [22]. For BP control, we recreated and adapted the model used to inform NICE guidance [23], identifying additional parameters through targeted searches and risk engines such as QRISK. We based the risky prescribing model on a previous model of NSAID prescribing used in a NICE osteoarthritis guideline (CG177). We adapted this model and populated it using results reported for three of the most commonly prescribed NSAIDs in the UK (diclofenac, naproxen, and ibuprofen) and aspirin. We created an additional model based on that used to inform the NICE Acute Kidney Injury guideline (CG169), covering NSAID prescribing in patients with CKD. Individual indicators were evaluated and their cost-effectiveness aggregated within each implementation package. We conducted a series of deterministic sensitivity analyses and a probabilistic sensitivity analysis. We used a willingness to pay threshold of £20,000 per QALY gained to indicate cost-effectiveness. We applied a 3.5% discount rate to costs and benefits post one year.

## Results

We assessed 278 practices for eligibility in January 2015, excluding 35 because of closure or earlier participation in the programme (Fig 1). We invited 243 (87%) practices to participate in February 2015; 56 (23%) opted out, largely because of workload pressures, and nine were excluded for other reasons.

In April 2015, we randomised 178 (73%) practices to Trial 1 (80 practices), Trial 2 (64 practices), or no intervention (34 practices). Analyses of the no intervention arm will be reported elsewhere. Within each trial, practices were randomised 1:1 to intervention arms.

Baseline characteristics for general practices and patients were well balanced by trial and intervention (Tables 1 and 2). The mean practice IMD was 30.22 (SD 13.86) within the top quarter of UK social deprivation [24].

No practices actively withdrew from the trial. One practice closed (BP intervention arm), another merged (risky prescribing) with a non-trial practice during the intervention period, and two trial practices merged (diabetes and risky prescribing). As outcome data for these practices were available up to the times of the second, third, and fourth feedback reports, respectively, all were included in the analysis using their most recent data.

Between May 2015 and April 2016, all intervention practices received feedback reports as intended, including an end-of-study report (S6 Table summarises intervention delivery). Educational outreach visits were delivered to 67 (47%) of 144 practices; 52 (68%) of the 77 practices declining visits gave no reason. Sixteen practices (24% of those receiving educational outreach visits) utilised additional pharmacist support. One hundred twenty-six (88%) practices joined the organisational groups, allowing them to access searches and computerised prompts (risky prescribing practices only). Second educational visits were delivered to eight (6%) practices.

**Table 1. General practice characteristics at baseline by trial arm.**

| Practice characteristics | Trial intervention | | | | Total (n = 144)[a] |
|---|---|---|---|---|---|
| | Diabetes control (n = 40) | Risky prescribing (n = 40) | BP control (n = 32) | Anticoagulation in AF (n = 32) | |
| List size | | | | | |
| Mean (SD) | 7,084.4 (3,786.5) | 7,175.8 (3,857.0) | 7,538.9 (4,932.9) | 7,421.3 (4,171.2) | 7,285.6 (4,128.9) |
| Overall QOF score 2014/2015[b] | | | | | |
| Mean (SD) | 535.4 (29.9) | 531.1 (35.8) | 527.2 (27.0) | 533.0 (21.7) | 531.9 (29.4) |
| Pre-intervention achievement (%) on primary outcome | | | | | |
| Diabetes control | | | | | |
| Mean (SD) | 32.9 (6.9) | 34.3 (7.7) | 32.5 (7.1) | 33.4 (5.5) | 33.3 (6.9) |
| Risky prescribing | | | | | |
| Mean (SD) | 7.9 (5.1) | 7.9 (3.6) | 7.3 (3.6) | 7.9 (2.5) | 7.8 (3.9) |
| BP control | | | | | |
| Mean (SD) | 66.5 (6.4) | 66.4 (7.0) | 65.9 (7.5) | 65.3 (6.2) | 66.1 (6.8) |
| Anticoagulation in AF | | | | | |
| Mean (SD) | 66.5 (14.4) | 67.3 (8.4) | 66.5 (10.8) | 66.3 (8.3) | 66.7 (10.8) |
| Deprivation score (IMD 2015) | | | | | |
| Mean (SD) | 30.3(13.0) | 32.4 (13.7) | 28.3 (14.6) | 29.3 (14.7) | 30.2 (13.9) |
| Number of GPs (FTE) | | | | | |
| Mean (SD) | 4.0 (3.0) | 4.1 (2.6) | 4.2 (3.1) | 4.0 (2.8) | 4.1 (2.8) |
| Number of GP partners (FTE) | | | | | |
| Mean (SD) | 3.3 (2.6) | 3.5 (2.4) | 3.6 (2.8) | 3.4 (2.5) | 3.4 (2.5) |
| Percentage of patients who would recommend practice to others | | | | | |
| Mean (SD) | 75.4 (14.5) | 73.9 (14.0) | 74.7 (13.9) | 76.0 (13.5) | 75.0 (13.9) |
| Percentage of patients who saw/spoke to nurse or GP within 48 hours of approach | | | | | |
| Mean (SD) | 51.7 (17.3) | 50.7 (15.6) | 50.8 (12.9) | 52.7 (16.1) | 51.5 (15.5) |
| Teaching practice? | | | | | |
| Yes | 14 (35%) | 15 (38%) | 13 (41%) | 15 (47%) | 57 (40%) |
| No | 26 (65%) | 25 (63%) | 19 (59%) | 17 (53%) | 87 (60%) |

[a]The practices randomised to the no intervention arm are not included here.

[b]There was one practice with a missing value for overall QOF score in Trial 2. The 2014/2015 QOF measured achievement against 81 indicators; practices scored points on the basis of achievement against each indicator, up to a maximum of 559.

Abbreviations: AF, atrial fibrillation; BP, blood pressure; FTE, full-time equivalent; GP, general practitioner; IMD, Index of Multiple Deprivation; QOF, Quality and Outcomes Framework, total score 2014/2015

**Table 2. Patient characteristics at baseline by trial arm (all patients from all practices in each intervention arm).**

| Patient characteristics | Trial intervention | | | | Total (n = 1,067,402)[a] |
|---|---|---|---|---|---|
| | Diabetes control (*n* = 288,130) | Risky prescribing (*n* = 290,407) | BP control (*n* = 249,571) | Anticoagulation in AF (*n* = 239,294) | |
| Age (years) | | | | | |
| Mean (SD) | 38.0 (22.9) | 37.6 (23.1) | 39.4 (23.2) | 39.0 (23.2) | 38.4 (23.1) |
| Sex[b] | | | | | |
| Female | 141,328 (49%) | 144,426 (50%) | 124,824 (50%) | 120,289 (50%) | 530,867 (50%) |
| Comorbidity[c] | | | | | |
| 0–3 | 276,280 (96%) | 277,184 (95%) | 239,455 (96%) | 229,329 (96%) | 1,022,248 (96%) |
| 4+ | 11,850 (4%) | 13,223 (5%) | 10,116 (4%) | 9,965 (4%) | 45,154 (4%) |

[a]The practices randomised to the no intervention arm are not included here.

[b]Each arm included <0.001% of patients defined as indeterminate or unknown.

[c]Measured as the number of QOF registers on which a patient appears.

Abbreviations: AF, atrial fibrillation; BP, blood pressure; QOF, Quality Outcomes Framework

Primary analyses demonstrated varying results across the implementation packages at 11 months post-randomisation (Table 3).

The diabetes implementation package had no observed effect on diabetes treatment targets. Achievement was 24.2% in intervention practices; 23.7% in control practices (adjusted odds ratio [OR] 1.03; 97.5% confidence interval [CI] 0.89–1.18; *p* = 0.693; ICC = 0.015).

**Table 3. Primary outcome achievement: Baseline rates, outcome rates, and ORs adjusted for baseline achievement and covariates.**

| Primary outcomes | Baseline achievement[a] (%) | Unadjusted model estimates | | | Adjusted model estimates[c] | | | |
|---|---|---|---|---|---|---|---|---|
| | | Outcome achievement[b] (%) | OR (97.5% CI) | *p*-value | Outcome achievement[b] (%) | OR (97.5% CI) | *p*-value | ICC |
| Trial 1: Diabetes control | | | | | | | | |
| Intervention | 33.7 | 24.3 | 1.016 (0.86–1.20) | 0.837 | 24.2 | 1.025 (0.89–1.18) | 0.693 | 0.015 |
| Control (risky prescribing) | 34.4 | 24.0 | | | 23.7 | | | |
| Trial 1: Risky prescribing | | | | | | | | |
| Intervention | 7.2 | 5.2 | 0.783 (0.59–1.04) | 0.052 | 4.9 | 0.815 (0.67–0.99) | 0.017 | 0.022 |
| Control (diabetes control) | 7.4 | 6.5 | | | 6.0 | | | |
| Trial 2: BP control | | | | | | | | |
| Intervention | 66.7 | 52.9 | 1.067 (0.94–1.22) | 0.266 | 53.6 | 1.053 (0.96–1.16) | 0.215 | 0.006 |
| Control (anticoagulation in AF) | 65.5 | 51.2 | | | 52.3 | | | |
| Trial 2: Anticoagulation in AF | | | | | | | | |
| Intervention | 66.4 | 72.8 | 0.866 (0.68–1.10) | 0.175 | 73.2 | 0.902 (0.75–1.09) | 0.214 | 0.009 |
| Control (BP control) | 67.5 | 75.5 | | | 75.2 | | | |

[a]Calculation of achievement for diabetes control and BP control at baseline uses any BP measurement taken in the previous 12 months.

[b]Calculation of achievement for diabetes control and BP control at outcome uses the most recent BP measurement taken.

[c]Variables controlled for in the adjusted analyses were as follows: patient-level sex and age, and practice-level baseline list size, CCG, pre-intervention achievement against primary outcomes, total QOF score 2014–2015, and proportion of patients with 0–3 comorbidities.

Abbreviations: AF, atrial fibrillation; BP, blood pressure; CCG, clinical commissioning group; CI, confidence interval; ICC, intra-cluster correlation coefficient; OR, odds ratio; QOF, Quality Outcomes Framework

**Table 4. Cost-effectiveness analysis: Mean probability sensitivity analysis outcomes at the practice level for the primary risky prescribing and BP control outcomes.**

| Probability sensitivity analysis outcomes | Risky prescribing | BP control |
|---|---|---|
| Mean incremental QALY | 0.90 | 13.00 |
| Mean incremental cost | £1,225 | £42,192 |
| ICER | £1,359 | £3,246 |
| Incremental Net Monetary Benefit* | £16,810 | £217,730 |
| Probability cost-effective | 0.79 | 0.52 |

*Assumes willingness to pay threshold of £20,000.

Abbreviations: BP, blood pressure; ICER Incremental cost-effectiveness ratio; QALY, quality-adjusted life year

The risky prescribing implementation package reduced high-risk NSAID and anti-platelet prescribing. In intervention practices, 4.9% of patients had a record of risky prescribing; 6.0% in control practices (adjusted OR 0.82; 97.5% CI 0.67–0.99; $p$ = 0.017; ICC = 0.022).

The BP implementation package had no observed effect on BP control. Achievement was 53.6% in intervention practices; 52.3% in control practices (adjusted OR 1.05, 97.5% CI 0.96–1.16; $p$ = 0.215; ICC = 0.006).

The AF implementation package had no observed effect on anticoagulant prescribing. Achievement was 73.2% in intervention practices; 75.2% in control practices (adjusted OR 0.90; 97.5% CI 0.75–1.09; $p$ = 0.214, ICC = 0.009).

There was little evidence of any intervention effects for any secondary outcome (S3–S6 Tables) except the risky prescribing implementation package, which showed some evidence of improvement against an individual indicator: patients aged 65 years or over prescribed aspirin and clopidogrel without co-prescription of gastro-protection. Adjusted prescribing levels were 25.3% in intervention practices, compared to 35.2% in control practices (adjusted OR 0.62; 97.5% CI 0.39–0.99; $p$ = 0.021). The conclusions for total serum cholesterol and HbA1c were robust to log transformation of the outcomes in a sensitivity analysis conducted following model diagnostic checks. We observed no intervention effects on any QOF indicators, including those not targeted by the implementation packages.

In estimating cost-effectiveness, we assumed equal costs for all four implementation packages. We conservatively doubled the cost per package. The total cost of package delivery was £175,592, and cost per practice was £2,439 (£175,592/144*2). Costs per patient (not only those eligible for indicator criteria) for an average practice list size of 7,130 were negligible (£0.28). No interventions were cost-saving. The risky prescribing package is highly likely to be cost-effective with an ICER of £1,359 and 79% chance of cost-effectiveness (Table 4), with results largely driven by two indicators. While the deterministic ICER for the BP package indicates cost-effectiveness, this is highly uncertain, with the probabilistic results indicating only a 52% chance of cost-effectiveness. We have not presented model results for the diabetes package as incremental costs and benefits were negligible, making model results unreliable. Results were robust to deterministic sensitivity analyses when, for example, we increased and reduced intervention costs. S1–S4 Figs present cost-effectiveness planes and acceptability curves for the risky prescribing and BP control implementation packages.

## Discussion

Our pragmatic, randomised trials found that an adaptable, multifaceted implementation package improved clinical care for only one of four high-impact indicators in general practices serving relatively socially deprived populations. The odds of risky prescribing for a patient in

an intervention practice were 18.5% lower than for a patient with the same characteristics in a control practice, which could ultimately be associated with changes in mortality, morbidity, and health service use, depending on generalisability to the general population. There was insufficient evidence of an effect upon diabetes control, BP control, and anticoagulation for AF. Our findings suggest that the design and delivery of implementation strategies need to account for differences in the nature of targeted clinical behaviours and go further than the kinds of adaptations in content that we applied.

Commonly used interventions to implement evidence-based practice, such as audit and feedback, educational outreach, and computerised prompts generally have modest if variable effects on clinical performance [12–14]. Tailoring such interventions to identified needs and barriers offers a means to enhance their effects, but how best to do this and improve patient outcomes remains uncertain [25].

Our findings confirm the effectiveness, generalisability, and value of interventions incorporating audit and feedback to improve prescribing safety in UK primary care [26–28]. The PINCER trial compared one-off feedback with pharmacist outreach visits to one-off feedback alone, whilst the D-QIP study provided financial incentives in addition to weekly feedback. Given our pragmatic approach to practice recruitment, our intervention and findings compare most closely to those of the EFIPPS trial, which found that quarterly feedback, with or without embedded behaviour change techniques, reduced risky prescribing by a similar magnitude [28]. Our modelling also indicated relative cost-effectiveness, mainly driven by improved levels of gastro-protection. It is noteworthy that the cost of the risky prescribing package was only £0.28 per patient.

We found no benefit of the implementation package on the management and outcomes of our targeted long-term conditions. Systematic review evidence suggests that interventions targeting systems of patient management along with patient-mediated interventions (which we did not include) are likely to be important components of strategies in this context [29].

The balanced incomplete block designs permitted comparison of intervention effects while mitigating any potential nonspecific performance effects of trial participation [30]. Our trials were pragmatic in three ways. Opt-out recruitment ensured that participating practices were generally representative of the wider population [19]. Data collection was minimally intrusive. For intervention delivery, all practices received but were not obliged to act on feedback reports, whilst outreach visits were optional. Hence, the implementation packages were tested under real-world conditions, increasing confidence in wider applicability to routine general practice settings. We further demonstrated no adverse impacts on incentivised indicators of care not targeted by the implementation package.

Our evaluation had five main limitations. First, the use of routinely collected data may have compromised the precision of trial outcomes and hence ability to demonstrate effects. However, we extracted structured data that are reasonably reliably coded in general practice, partly incentivised by QOF. Second, given the multifaceted nature of the implementation package, we cannot make any inferences about the relative effects of individual intervention components. Third, educational outreach visits were delivered by facilitators not allocated to specific trial arms, thereby risking contamination between arms. We had instructed facilitators to focus only on delivering the implementation package to which each practice was assigned. Fourth, an 11-month follow-up period may have been too short to detect changes in patient outcomes, such as those related to diabetes or BP control, especially if a number of general practices only received educational outreach visits later in this period. This explanation is unlikely, as other trials have demonstrated changed clinical outcomes within similar durations of follow-up [29]; we also did not detect any improvements in processes of care for diabetes and BP control, respectively. Fifth, our composite endpoint for diabetes control requiring

achievement of all HbA1c, BP, and cholesterol treatment goals may have been too demanding. Our clinical and patient advisors had considered this endpoint fair, if challenging.

Our work highlights three methodological issues. First, the implementation package effect on risky prescribing was modest but important at a population level. Foregoing a randomised design, as has been suggested for quality improvement research [31], would have reduced confidence in the validity of our findings and risked false positive conclusions. For example, the trial design accounted for temporal changes, including improvements in risky prescribing and anticoagulation achievement and declines in BP and diabetes achievement (Table 3), which would otherwise be difficult to interpret. Second, whilst we see the pragmatic design as a strength, a more explanatory approach could have made full engagement with our implementation package a condition of trial participation. Such mandating is seldom possible or even desirable in quality improvement programmes dependent upon professional engagement, particularly if they encourage 'gaming' behaviours to achieve goals whilst circumventing real action. Similarly, our opt-out approach to recruitment may have reduced self-selection of more enthusiastic practices, as well as administrative burden. Those responsible for leading quality improvement initiatives often specifically wish to include less enthusiastic or poorer performing practices. Third, a critical challenge prior to pragmatic evaluations is to develop interventions that are sufficiently durable to withstand the relatively hard-pressed and evolving environments of clinical practice. Whilst we followed the UK Medical Research Council framework for the development and evaluation of complex interventions, practice engagement with our implementation package was highly variable [32]. We would now recommend more intensive field work involving iterative cycles of testing and refining interventions prior to scaling up for definitive evaluation.

We had set out to develop an implementation package that could be adapted for different clinical priorities in primary care. We offer four interrelated explanations for the observed differences in intervention effects. First, the targeted clinical behaviour(s) and associated endpoints varied according to the extent of control held over them by clinical staff. Clinicians could make relatively straightforward changes to reduce risky prescribing, such as adding gastro-protection for prescribed aspirin, with limited input from patients. The observed significant difference in achievement of the associated secondary outcome on gastro-protection provides evidence to support such a mechanism (although we would still advise cautious interpretation, given that these analyses were not formally adjusted as we tested across multiple secondary endpoints). In contrast, improving BP control can require at least two consultations and changes in patient behaviour, as well as finding a pharmacological agent that is acceptable to and effective for an individual patient. This is consistent with evidence that adherence to clinical recommendations that are more complex or disruptive to routine practice is lower compared with simpler recommendations [33]. Interventions involving audit and feedback have also been shown to reduce other discouraged prescribing behaviours, namely of antibiotics [34], and offer a means to address urgent priorities such as rising opioid prescribing [35]. Second, the risky prescribing package included an automated computerised prompt requiring a one-step justification for continued prescribing. Such prompts are generally effective in changing prescribing behaviour [14]. However, a recent UK trial found that on-screen reminders did not improve adherence to the other prescribing behaviour we targeted, anticoagulation prescribing for AF [36]. With the benefit of hindsight, this is unsurprising. Qualitative work we undertook before the trials revealed that anticoagulation prescribing comprises a series of deceptively complex considerations and behaviours, which include balancing benefits and risks with patients [15]. Clinicians sometimes lacked confidence in starting treatment, given that they encountered it relatively infrequently in routine practice and felt frustrated by complicated guidance that made treatment difficult to explain to patients. Third, the number

of patients requiring action to achieve indicators varied by trial arm, with far more in the diabetes and BP control arms relative to risky prescribing and anticoagulation for AF (see S5 Table). Hence, the larger numbers of patients requiring action may have undermined clinicians' perceived feasibility and motivation. Moreover, making changes for even a small number of patients within the risky prescribing indicator set would have had a comparatively larger effect upon achievement relative to other indicators. Fourth, the indicators for three targeted long-term conditions were incentivised in the QOF. As well as aligning the implementation package with existing quality improvement schemes, our intention had been to encourage practices to move beyond the QOF by adopting more challenging, evidence-based goals. Within the present constraints of UK primary care, it was hard for practices to take on additional, unrewarded work. Clinicians may, nevertheless, have been motivated to address risky prescribing because it concerned patient safety.

In conclusion, an adaptable implementation package improved prescribing safety in general practice. However, we observed no benefits of the package within the context of an existing financial incentive system targeting similar aspects of care for three long-term conditions. Improving patient outcomes for long-term conditions requiring relatively complex management may require systemised approaches that target patient as well as professional behaviour.

## Supporting information

**S1 Table. Secondary outcomes from Trial 1: Achievement of individual indicators that contributed to composite outcomes; processes of care; and continuous intermediate clinical outcomes. All adjusted for covariates and baseline achievement of primary outcomes. Table presents mean percentage achievement, unless otherwise stated.** Variables controlled for in the adjusted analyses were as follows: patient-level sex and age, and practice-level baseline list size, CCG, pre-intervention achievement against primary outcomes, total QOF score 2014–2015, and proportion of patients with 0–3 comorbidities. ACE-I, angiotensin-converting-enzyme inhibitor; ACR: albumin:creatinine ratio; ARB, angiotensin receptor blocker; BMI, body mass index; BP, blood pressure; CCG, clinical commissioning group; CI, confidence interval; CKD, chronic kidney disease; eGFR, estimated glomerular filtration rate; HbA1c, haemoglobin A1c; NSAID, non-steroidal anti-inflammatory drug; PCR, protein:creatinine ratio; QOF, Quality Outcomes Framework.
(DOCX)

**S2 Table. Secondary outcomes from Trial 1: Achievement of QOF indicators relating to the implementation packages; and non-trial related QOF indicators. All adjusted for covariates and baseline achievement of primary outcomes.** Note: Formal statistical testing was inappropriate due to violation of the modelling assumptions for the following trial-related indicators: DM006, DM014, DM018; and the following non-trial related indicators: CHD005, CHD007, MH002, MH003, SMOK004, and SMOK005. Summary statistics only are presented for these indicators. [†]The HbA1c and total serum cholesterol continuous intermediate clinical outcomes were analysed using a log transformation in order to satisfy the modelling assumptions. The predicted means presented are on the untransformed (original) scale, but the estimated intervention effect (and 97.5% CI) are on the log scale. [*]If urine albumin:creatinine ratio ≥3, or retinopathy, or record of cerebrovascular accident or transient ischemic attack. Variables controlled for in the adjusted analyses were as follows: practice-level baseline list size, CCG, pre-intervention achievement against primary outcomes, and total QOF score 2014–2015. ACE-I, angiotensin-converting-enzyme inhibitor; ARB, angiotensin receptor blocker; CCG, clinical commissioning group; CHD, coronary heart disease; CI, confidence interval; CKD, chronic kidney disease (stage 3–5); COPD, chronic obstructive pulmonary

disease; HbA1c, haemoglobin A1c; IFCC, International Federation of Clinical Chemistry and Laboratory Medicine; PAD, peripheral arterial disease; QOF, quality and outcomes framework; RCP, Royal College of Physicians; TIA, transient ischemic attack.
(DOCX)

**S3 Table. Secondary outcomes from Trial 2: Achievement of individual indicators that contributed to composite outcomes; processes of care; and continuous intermediate clinical outcomes. All adjusted for covariates and baseline achievement of primary outcomes. Values are percentage achievement, unless otherwise stated.** Variables controlled for in the adjusted analyses were as follows: patient-level sex and age, and practice-level baseline list size, CCG, pre-intervention achievement against primary outcomes, total QOF score 2014–2015, and proportion of patients with 0–3 comorbidities. *If urine albumin:creatinine ratio $\geq 3$, or retinopathy, or record of cerebrovascular accident or transient ischemic attack. AF, atrial fibrillation; BP, blood pressure; CCG, clinical commissioning group; $CHA_2DS_2$-VASc, congestive heart failure, hypertension, age>75, diabetes, stroke, vascular disease, age between 65 and 74, and female sex; CHD, coronary heart disease; CI, confidence interval; CKD, chronic kidney disease; CVD, cardiovascular disease; HTN, hypertension; PAD, peripheral arterial disease; QOF, Quality Outcomes Framework; TIA, transient ischemic attack.
(DOCX)

**S4 Table. Secondary outcomes from Trial 2: Achievement of QOF indicators relating to the implementation packages; and non-trial-related QOF indicators. All adjusted for covariates and baseline achievement of primary outcomes.** Note: Formal statistical testing was inappropriate due to violation of the modelling assumptions for the following trial-related indicators: AF006, AF007; and the following non-trial related indicators: CHD005, CHD007, MH002, MH003, SMOK004, and SMOK005. Summary statistics only are presented for these indicators. Variables controlled for in the adjusted analyses were as follows: practice-level baseline list size, CCG, pre-intervention achievement against primary outcomes and total QOF score 2014–2015. CCG, clinical commissioning group; $CHA_2DS_2$-VASc, congestive heart failure, hypertension, age>75, diabetes, stroke, vascular disease, age between 65 and 74, and female sex; CHD, coronary heart disease; CI, confidence interval; CKD, chronic kidney disease; COPD, chronic obstructive pulmonary disease; PAD, peripheral arterial disease; QOF, Quality and outcomes framework; RCP, Royal College of Physicians; TIA, transient ischemic attack.
(DOCX)

**S5 Table. Key sample size assumptions.** Data from an earlier work package of the ASPIRE programme were used to inform the trial sample size assumptions. Mean cluster size, cluster size coefficient of variation, ICC, and mean achievement rates were calculated using real data from practices within West Yorkshire for each primary outcome indicator. [a]Mean achievement is the control arm achievement rate for each primary outcome indicator estimated using data available from the earlier work package. ICC, intra-cluster correlation coefficient.
(DOCX)

**S6 Table. Intervention delivery across trial practices.** [a]One practice in the risky prescribing arm merged with a non-ASPIRE practice in advance of the final feedback report. [b]One practice in the BP arm closed in advance of the third feedback report. [c]Only practices receiving an initial outreach visit were offered additional support; these practices are used as the denominator in the percentages presented. [d]These granted access to the computerised searches (all arms) and prompts (risky prescribing only). BP, blood pressure.
(DOCX)

**S1 Fig. Scatterplot of simulated incremental cost and QALY for the composite indicator of risky prescribing.** ICER, incremental cost-effectiveness ratio; PSA, probability sensitivity analysis; QALY, quality-adjusted life year.
(TIF)

**S2 Fig. Cost-effectiveness acceptability curve for the composite indicator of risky prescribing.** QALY, quality-adjusted life year; WTP, Willingness to Pay.
(TIF)

**S3 Fig. Scatterplot of simulated incremental cost and QALY for the composite indicator of BP control.** BP, blood pressure; ICER, incremental cost-effectiveness ratio; PSA, probability sensitivity analysis; QALY, quality-adjusted life year.
(TIF)

**S4 Fig. Cost-effectiveness acceptability curve for the composite indicator of BP control.** BP, blood pressure; QALY, quality-adjusted life year; WTP, Willingness to Pay.
(TIF)

**S1 CONSORT Checklist.**
(DOCX)

**S1 Statistical Analysis Plan.**
(DOCX)

**S1 Trial Protocol.**
(DOC)

## Acknowledgments

The ASPIRE programme team comprises Susan Clamp, Rebecca Lawton, Rosie McEachan, Martin Rathfelder, Judith Richardson, Tim Stokes, Vicky Ward, Robert West, and Ian Watt, in addition to the named authors. The ASPIRE programme team can be contacted via Robbie Foy (r.foy@leeds.ac.uk). We acknowledge the contributions of Andrew Davies, Peter Heudtlass, Chris Jackson, and John Turgoose and thank the members of the ASPIRE Programme Steering Committee for their advice and expertise. We thank Richard Neal, Suzanne Richards, and Noah Ivers for helpful comments on earlier versions of this manuscript.

**Disclaimer:** The views expressed are those of the authors and not necessarily those of the NIHR or the Department of Health and Social Care.

## Author Contributions

**Conceptualization:** Thomas A. Willis, Liz Glidewell, Amanda J. Farrin, Claire Hulme, Suzanne Hartley, Robbie Foy.

**Data curation:** Michelle Collinson, Michael Holland.

**Formal analysis:** Michelle Collinson, Amanda J. Farrin, Michael Holland, David Meads, Armando Vargas-Palacios.

**Funding acquisition:** Liz Glidewell, Amanda J. Farrin, Claire Hulme, Suzanne Hartley, Robbie Foy.

**Investigation:** Thomas A. Willis, Liz Glidewell, Amanda J. Farrin, Duncan Petty, Sarah Alderson, Robbie Foy.

**Methodology:** Thomas A. Willis, Michelle Collinson, Liz Glidewell, Amanda J. Farrin, David Meads, Duncan Petty, Sarah Alderson, Suzanne Hartley, Paul Carder, Stella Johnson, Robbie Foy.

**Project administration:** Thomas A. Willis, Liz Glidewell, Duncan Petty, Suzanne Hartley, Paul Carder, Stella Johnson.

**Resources:** Paul Carder.

**Supervision:** Amanda J. Farrin, Robbie Foy.

**Writing – original draft:** Thomas A. Willis, Michelle Collinson, Michael Holland, David Meads, Robbie Foy.

**Writing – review & editing:** Thomas A. Willis, Michelle Collinson, Liz Glidewell, Amanda J. Farrin, Michael Holland, David Meads, Claire Hulme, Duncan Petty, Sarah Alderson, Suzanne Hartley, Armando Vargas-Palacios, Paul Carder, Stella Johnson, Robbie Foy.

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
