## [Decision Letter · Decision Letter 0]

29 Oct 2019

Dear Dr. Willis,

Thank you very much for submitting your manuscript "An adaptable implementation package targeting evidence-based indicators in primary care: a pragmatic cluster-randomised evaluation" (PMEDICINE-D-19-02682) for consideration at PLOS Medicine. 

Your paper was evaluated by a senior editor and discussed among all the editors here. It was also discussed with an academic editor with relevant expertise, and sent to three independent reviewers, including a statistical reviewer. The reviews are appended at the bottom of this email and any accompanying reviewer attachments can be seen via the link below:

[LINK]

In light of these reviews, I am afraid that we will not be able to accept the manuscript for publication in the journal in its current form, but we would like to consider a revised version that addresses the reviewers' and editors' comments. Obviously we cannot make any decision about publication until we have seen the revised manuscript and your response, and we plan to seek re-review by one or more of the reviewers. 

We expect to receive your revised manuscript by Nov 19 2019 11:59PM. Please email us (plosmedicine@plos.org) if you have any questions or concerns.

We look forward to receiving your revised manuscript. 

Sincerely,

Caitlin Moyer, Ph.D.

Associate Editor 

PLOS Medicine

plosmedicine.org

1. Competing Interest Statement: Please clarify the role that DP, PC and SJ had in the implementation and analyses of the study, in light of their associations with NHS Bradford Districts CCG, and Prescribing Support Services, Ltd.

2. Data Availability Statement: PLOS Medicine requires that the de-identified data underlying the specific results in a published article be made available, without restrictions on access, in a public repository or as Supporting Information at the time of article publication, provided it is legal and ethical to do so. Please see the policy at 

http://journals.plos.org/plosmedicine/s/data-availability

and FAQs at http://journals.plos.org/plosmedicine/s/data-availability#loc-faqs-for-data-policy. 

Please include an appropriate contact (web and/or email address) for inquiries to obtain access to the data (please note: this cannot be a study author).

3. Abstract Background: The final sentence should clearly state the study question/hypothesis to be tested.

4. Abstract: Methods and Findings: Please define the control for the intervention.

5. Abstract: Methods and Findings (Line 120): You mention here that 34 practices were recruited as non-intervention controls. This could be misleading because you do not include any analyses of these controls in this manuscript, and comparisons are made to within-arm controls. Please remove the mention of these non-intervention controls here, and clarify the control for these analyses.

6. Abstract: Methods and Findings: In the last sentence of the Abstract Methods and Findings section, please describe the main limitation(s) of the study's methodology.

7. Abstract: Conclusions: Please interpret the study based on the results presented in the abstract, emphasizing what is new without overstating your conclusions; the phrase "In this study, we observed ..." may be useful. Please avoid vague statements such as "these results have major implications for policy/clinical care". Mention only specific implications substantiated by the results. Specifically, both sentences of the conclusion are vague and do not clearly reflect what can be concluded from the study findings.

8. Author Summary: Thank you for including an Author Summary. The Author Summary should immediately follow the Abstract in your revised manuscript. 

9. Author Summary: “What do these findings mean?”: Please revise the final point to: “In this study, we found that a ‘one-size-fits-all’ strategy did not work…” Please also revise this point to clarify what is meant by “did not work” and “the nature of clinical behaviors targeted”. 

10.Introduction: Please conclude the Introduction with a clear description of the study question or hypothesis.

11. Methods: The following points in the Methods differ from those described in the trial registry (ISRCTN registry ISRCTN91989345): In manuscript: “Audit and Feedback”, “Educational Outreach”, and “Prompts and Reminders” are described (Methods: Procedures: Lines 224-249). In the registered trial protocol, “GP Appraisal and Revalidation” and “Patient Mediated Intervention” are also described. Please explain, including in the manuscript.

12. Methods: The following points in the Methods differ from those described in the trial registry: There are a number of primary and secondary outcomes described in the manuscript (Methods: Outcomes: Lines 253-312). However, the registered protocol describes the primary outcome as: “Adherence; Timepoint(s): Adherence will be measures up to 12 mth post randomisation”, with no secondary outcomes described. Please clarify and explain the discrepancy. If the outcomes were not pre-specified in the protocol, please indicate that they were post hoc and explain why they were added. Post hoc comparisons should be presented as hypothesis generating rather than conclusive. Please explain, including in the manuscript.

13. Methods: Line 328: Please provide a link to the pre-specified statistical analysis plan referred to here, and include as supplementary information.

14. Results: Line 347: Please provide statistical evidence (in the appropriate tables) for the statement that: “Baseline characteristics for general practices and patients were well balanced by trial and intervention (Tables 1 and 2).”

15. Results: Line 375: Please provide a reference for the statement: “The mean practice IMD was 30.22 (SD 13.86), within the top quarter of UK social deprivation.”

16. Results: Lines 429-441: For the following adjusted analyses (the main analyses describing effects of diabetes, risky prescribing, BP, and AF implementation package on outcomes) please also provide the unadjusted analyses.

17. Results: Lines 445-447: For the following adjusted analyses (the analysis of the effect of risky prescribing implementation on adjusted prescribing levels) please also provide the unadjusted analyses.

18. Results: Lines 462-463: Please revise to: “Supplementary Figures 1-4 present cost-effectiveness planes and acceptability curves for the risky prescribing and blood pressure control implementation packages.” or similar.

19. Discussion: Line 541-543: Please provide an in-text reference to Supplementary Table 1 (S1 Table) for the statement that there are fewer eligible patients in the risky prescribing/anticoagulation AF arms.

20. Discussion: Please present and organize the Discussion as follows: a short, clear summary of the article's findings; what the study adds to existing research and where and why the results may differ from previous research; strengths and limitations of the study; implications and next steps for research, clinical practice, and/or public policy; one-paragraph conclusion.

21. Discussion: Thank you for including the link to the ASPIRE protocol, found at the end of the Discussion, after the Author Summary. It may be helpful to refer to any prospective study/analysis protocol in the Methods section of the manuscript. Also, the link does not seem to work. Please include the study protocol document and analysis plan, with any amendments, as Supporting Information to be published with the manuscript if accepted.

22. Figure 1: In the flow diagram, please indicate the number of individuals in each group analyzed in the ITT analysis. Please make it more clear in the diagram that the non-intervention control group (n=34) were not analyzed in the analyses being described here. This is written in the figure legend, but please include some visual indication in the diagram (text and/or shading) to make this point clear to the reader.

23. Figure 1: Please define the abbreviations: A&F (follow up).

24. Supplementary Figures 1, 2, 3, and 4 (S1, S2, S3, S4 Figures): Please define the following abbreviations: ICER, PSA, QALY, WTP.

25. Table 3, and Supplementary Tables 3, 4, 5, and 6: Please provide the unadjusted comparisons as well as the adjusted comparisons.

26. Table 3, and Supplementary Tables 3, 4, 5, and 6: Please specify the variables controlled for in the adjusted analyses in the Table legend.

27. Table 3: Please define the following abbreviations: CI, ICC.

28. Supplementary Table 1: Please clarify sources of data in support of assumptions. Please explain the “mean achievement” measure in the table legend.

29. Supplementary Table 3 :Please define the following abbreviations: CI, BP, HbA1c, ACR, PCR, eGFR, BMI, NSAID, ACE-I, ARB, CKD.

30. Supplementary Table 4: Please define the following abbreviations: CI, QOF, IFCC-HbA1c, RCP, CHD, PAD, TIA, COPD.

31. Supplementary Table 5: Please define the following abbreviations: CI, BP, HTN, CKD, CHD, PAD, TIA, CVD, AF, CHA(2)DS(2)-VASc.

32. Supplementary Table 6: Please define the following abbreviations: CI, QOF, CHA(2)DS(2)-VASc, RCP.

33. Checklist: Please complete the CONSORT checklist and ensure that all components of CONSORT are present in the manuscript. Please include the completed checklist as Supporting Information (e.g. S1 Checklist). When completing the checklist, please use section and paragraph numbers, rather than page numbers.

Comments from the reviewers:

Reviewer #1: Alex McConnachie

Willis et al report on two cluster randomised trials of implementation packages delivered to general practices to improve diabetes control, risky prescribing, blood pressure control, and anticoagulation in AF. This review looks at the use of statistics in the paper.

Overall, the trial design, statistical methods, and presentation of results are very good. My one minor point is that it appears that the primary results were judged at a significance level of 2.5%; this is not explained in the statistical methods, but is mentioned in the results. I presume the reason for this is that each trial involved the assessment of two interventions, so a 2.5% significance level was use to preserve an overall 5% Type I error rate. This should be explained in more detail in the methods section.

However, it could also be argued that the significance level should be set at 1.25%, since the trial as a whole involved testing the effect of four interventions. On that basis, the effect of the risky prescribing intervention would not be considered statistically significant. Either way, with multiple primary analyses, a clearer description of how the authors controlled the false discovery rate should be given.

Was there a statistical analysis plan? Has this been published, or should it be provided as a supplement?

Reviewer #2: This paper presents a trial that evaluates implementation science interventions for multiple evidence based indicators. I agree with the authors that this approach could be useful in the context of multiple morbidity.

However, the paper includes a number of flaws. The main ones are: the aim or hypothesis is not stated; the nature of the 'adaptation' included in the intervention is not sufficiently specified; the statistical results are not correctly interpreted; consequently, the conclusions may not be justified.

Abstract: This could be clearer. In particular the sentence on randomisation would naturally follow-on from the sentence on recruitment.

Introduction: This is clear and succinct. More could have been made of multiple morbidity. The study appears to lack a clear aim or hypothesis. The final paragraph here, explains what was done but not the purpose.

Methods: The study design could be explained more clearly. Initially, it says that this is 'two parallel cluster randomised trials using balanced incomplete block designs'. This should be explained more completely: what are the intervention and comparator trial arms in each trial. Why does this represent a balanced incomplete block design?

Later it refers to 'two stage randomisation' which suggests a single trial , the written description is not clear though Figure 1 is. Looking at Figure 1, the design is clear but the research question is not.

It appears that the no intervention control trial arm was not included in the protocol but was only added because recruitment exceeded expectations. The paper then says that the control group analysis will not be reported in this study. It may be best to focus this paper on what was in the protocol only.

The software for the minimisation could be mentioned. Also, how allocation was concealed as distinct from blinding.

The description of the intervention is generic. It is not made clear how the intervention package differed for each of the four trial arms. What are the comparisons being made? The title refers to an 'adaptable' intervention though the basis of adaptation is not mentioned in the methods.

Under outcomes it refers to 'primary outcomes for each intervention arm'. Normally there is one primary outcome for a trial. Here there are four intervention arms, but the study comprises two trials.

Multiple comparisons require additional discussion.

Missing data: initially it says that missing data could not be assessed, then it says that data were assumed missing at random. Does it mean that not recorded values were assumed to be not performed for process items? Was imputation performed for values like blood pressure? or were complete case analyses performed it is not clear.

Results: 

Page 24 lines 429 and 436 and 439, where it says 'no effect', 'absence of evidence is not evidence of absence' as Doug Altman pointed out. https://www.bmj.com/content/311/7003/485

line 434, where it says 'statistically significant improvement. The paper should interpret P values following the ASA recommendations. Cut points such as P<0.05 or P<0.025 should not be used. Reporting should be based on ASA guidelines with not 'bright lines' for interpretation such as P<0.025. https://amstat.tandfonline.com/doi/full/10.1080/00031305.2016.1154108#.XVpvSuhKiUk

Similarly line 442 where it says 'no statistically significant effect'

Line 443 where it says 'significantly improved one of the indicators', was adjustment made for multiple comparisons?

In genereal insufficient details are provided for the cost-effectiveness analysis.

Line 458 'the deterministic ICER for the BP packages indicates cost-effectiveness'. How can this be so when the intervention had 'no effect' (line 436).

Discussion 

Again, the paper concludes that the intervention package had no effect for three out of four indicators, but this could be an 'absence of evidence is not evidence of absence error'. Consequently, the conclusion may be too strong. Perhaps more evidence will show that an adaptive approach may be effective.

The paper arrives at the conclusion that an adaptable intervention 'does not work' but it is not made clear from the analysis plan how the various comparisons that were made could lead to such a conclusion. 

Reviewer #3: This manuscript describes the findings from a large implementation trial intended to improve quality in primary care for 4 common conditions. The authors went through a careful process of selecting performance measures that matter and designing an implementation package based on the best available evidence at the time. They then implemented this in a large number of GP practices, and assessed the fidelity of the intervention. No differences were seen in their primary outcomes except a decrease in what they term "risky prescribing", meaning a patient safety issue, from 6.0% to 4.9%. A cost effectiveness analysis showed that for this one target, their intervention was cost-effective. The authors conclude that "a one size-fits-all implementation package only worked for 1 in 4 indicators".

The topic area of this manuscript - improving quality and safety in primary care - is high on the agenda for all health care systems. As someone who regularly reviews quality improvement publications as part of their scholarly activities, I can say with authority that this manuscript is a very high outlier in terms of the preparation and planning that went into the intervention and the way it is reported. 

Note I am not a cost-effectiveness analyst and so can make no informed critique of their methods or results in this regard.

The results of the QI process must have come as a big disappointment to the investigators, and will be disappointing to readers as well, but these are results that have to be widely disseminated in order for ongoing and planned implementations to be changed now, to avoid continuing ineffective QI initiatives. 

I have only a few comments for the authors to consider, all in the discussion sections about "Why?" The first and most obvious is already touched on briefly by the authors, namely that of their primary outcomes, the risky prescribing measure is the easiest for a GP to do. For two other conditions the performance measure is an intermediate outcome (or even a composite of several intermediate outcomes), and therefore much harder for the GP to influence. The AF indicator is a process measure but one that requires 1) that the GP calculate the CHADS-VASC score (maybe the EHR already does this automatically - if so then the manuscript should make this point) AND 2) prescribe a medication known to have a certain risk to it, plus be a pain-in-the-rear-end for patients and practices to monitor: they are signing themselves up to regular monitoring of INR, unless they are going with a new oral agent, which i suspect at the time this trial was going would have been rare in UK GP practices. As opposed to the "risky prescribing" measure, which only required someone to choose paracetamol instead of an NSAID or add omeprazole if they go with the NSAID. So, I wonder if the authors want to discuss a little more the 'easiness' of meeting this measure as opposed to the others. I understand that the process measures for diabetes - also easy things to do - didn't have significant improvements in their intervention group (although they all were a little better in the intervention vs the control), but most of them are topped out, anyway, with not a lot of room for improvement. So I think there is more-than-a-hint that these data are compatible with the conclusion that influencing a process measure is easier than influencing an outcome measure. Perhaps the authors want to expand a little on this text in the discussion, right now it is two sentences and, to me, focuses on the wrong thing- namely that BP control requires 2 visits - instead of the much-more-difficult problem of finding a pharmaceutical that is both effective and acceptable to the patient. The current text also does not deal with the AF performance measure, which like the risky prescribing measure only requires the GP to add a medication. Why isn't adding warfarin as easy as adding omeprazole? Understanding why is key to understanding why their intervention had an effect for one and did not for the other.

Secondly, I wonder whether the "patient safety" angle of the "risky prescribing" measure played a role. The other measures are all clinical quality, whereas the risky prescribing measure is a patient safety issue, making it perhaps more important in the GP's eyes.

Thirdly, I think a little more information about the EHR prompt is warranted. Have I concluded correctly that this was the only measure that had an EHR prompt included? And was the prompt a hard stop, requiring an over-ride, or a soft-stop, something easily blown through by the GP? Judging by the size of the effect, I am assuming a soft stop. These data are also consistent with a conclusion that EHR prompts work for simple actions, which the authors note, but some more information about the prompt, and why there wasn't a prompt for the warfarin measure, would help readers understand when a prompt might work (I am presuming this is because the NSAID prompt was triggered by the e-prescribing of an NSAID, which the EHR can then assess against age, concomitant cytoprotection, GFR, and concomitant warfarin, whereas the AF measures would require an alert to be triggered by something else much more complex. The authors cite another trial in this regard, but rather than make readers go look up that trial they would benefit from a sentence or two about what makes one easy and the other complex).

Lastly, there is the question of additional incentives. When practices are already being paid via QOF for doing a host of things, how much mental energy is left over to participate in something where they aren't getting more payments for doing the "thing"?

So...bottom line, an important study with important (but disappointing) findings, that in my view would benefit from a couple hundred extra words from the authors speculating about why they think some aspects did and did not work as expected.

-paul shekelle

[LINK]

---

## [Decision Letter · Decision Letter 1]

14 Jan 2020

Dear Dr. Willis,

Thank you very much for re-submitting your manuscript "An adaptable implementation package targeting evidence-based indicators in primary care: a pragmatic cluster-randomised evaluation" (PMEDICINE-D-19-02682R1) for review by PLOS Medicine.

I have discussed the paper with my colleagues and the academic editor and it was also seen again by 3 reviewers. I am pleased to say that provided the remaining editorial and production issues are dealt with we are planning to accept the paper for publication in the journal.

[LINK]

We look forward to receiving the revised manuscript by Jan 21 2020 11:59PM. 

Sincerely,

Caitlin Moyer, Ph.D.

Associate Editor 

PLOS Medicine

plosmedicine.org

Requests from Editors:

1. Competing Interest Statement: Thank you for clarifying the roles of the authors in the updated competing interest section in your manuscript. Please update the competing interest statement on the manuscript submission form. 

2. Data Availability Statement: Thank you for providing the contact information for access to your study data. We suggest you shorten the Data Availability Statement to: “Data cannot be shared publicly owing to a need to main patient confidentiality. Interested researchers may contact ctru-dataaccess@leeds.ac.uk to request and obtain relevant data.” Please update this in the manuscript submission form with your revised submission.

3. Throughout: Abstract Lines 104 and 112, Introduction Line 198, and Methods Line 239: Please replace the term “usual practice” with “implementation control”, as your control comparison condition is not accurately described as usual practice.

4. Throughout:Trademark (™) symbol: Please remove the trademark symbol throughout the manuscript (Lines 218, 309, 363, 367, 396).

5. Abstract: Line 104-105: In the final sentence, please clarify the nature of the indicators (indicators of what?)

6. Abstract: Line 108-109: Please revise this sentence to: “We used ‘opt-out’ recruitment, and practices that did not opt out were randomly assigned to to an implementation package…”

7. Abstract: Line 123-124: Please describe the confounds for which you adjusted.

8. Abstract: Line 127: Rather than saying that the implementation package “had no effect on other primary endpoints”, please provide the adjusted ORs ( with CIs and p values) for all four primary comparisons from Table 3.

9. Abstract: Conclusions: Your conclusion only touches on the cost-effectiveness of the intervention. Can you please include your main conclusions regarding the primary endpoint (i.e. adherence is your registered primary endpoint, and the description above in the abstract describes specific patient-level primary endpoints that could be discussed). At Line 135, please avoid using the term “relatively cost-effective”; instead, please use quantitative terms to describe cost-effectiveness.

10. Author Summary: “What did the researchers do and find?”: Please remove the word “highly” from the first bullet point. Also, please clarify if you are reporting results of one trial or two trials, and mention the outcomes being assessed.

11. Author Summary: “What do these findings mean?”: In the first point, we suggest you also mention your conclusion regarding the primary outcome of the study, in addition to mentioning the cost-effectiveness outcome.

12. Methods: Please add the following statement to the Methods section: “This study is reported as per the Consolidated Standards of Reporting Trials (CONSORT) guideline for cluster randomized trials (S1 Checklist).”

13. Discussion: Lines 532-533: Please clarify the second part of this sentence, we suggest: “...which could ultimately be associated with changes in mortality, morbidity and health service use, depending on generalizability in the general population.” or similar.

14. Discussion: Lines 550-551: Please clarify this sentence: “The risky prescribing package cost of £0.28 per patient is inexpensive.” Inexpensive relative to what?

15. Discussion: Line 558: Please remove the word “highly” from the sentence.

16. References: Please double check that all references are formatted with the "Vancouver" style for reference formatting, and see our website for other reference guidelines https://journals.plos.org/plosmedicine/s/submission-guidelines#loc-references

For example, reference 7 should be “PLoS ONE”, ref 18, 28, 32, are not complete/have extraneous text; ref 21, 27, should be N Engl J Med.

17. Link to Trial Protocol: Line 649-650: Thank you for including the trial protocol with your supporting information. The Supporting Information will be available online with the manuscript. As you have included the protocol document, you can remove the extra link here.

18. Table 2: Please check the “a” “b” “c” superscripts in the table footnote, there appears to be some discrepancy.

19. Supplementary Figure 2 and 4: Please integrate the “thousands” label for the X axis units with the X axis legend.

20. Checklist: Thank you for including your CONSORT checklist. For Item 1a, please change the location to “Title” and for item 1b, please change the location to “Abstract”. For item 6b, you could direct the reader to where you discuss changing your primary outcome assessment timepoint to 11 months rather than 12 months. For item 24, please reference your trial protocol included in the supporting information (e.g. S1 text). For item 25, please reference “Funding” as the section for this information. 

Comments from Reviewers:

Reviewer #1: Having reread the paper, and the authors' responses to my and the other reviewers' comments, I am happy with the changes that have been made. I have no further comments to make.

Reviewer #2: The revision has generally responded to the review comments.

In the Abstract where it reads: 'The implementation package reduced risky prescribing (odds ratio 0.82; 97.5% confidence interval (CI) 0.67 to 126 0.99, p = 0.017) with an incremental cost effectiveness ratio of £1,359 per quality-adjusted life year but had no effect on other primary endpoints. No statistically significant effects were observed in any secondary outcome except for reduced co-prescription of aspirin and clopidogrel without gastro-protection in patient aged 65 and over (adjusted OR 0.62; 97.5% 130 CI 0.39 to 0.99; p = 0.021). '

It appears that the abstract is reporting only those results that are associated with P values less than 0.025, whereas the other outcomes are referred to as having 'no significant effects'. It would be preferable to report each of the primary outcomes in the Abstract.

The ASA guidelines on P values make the recommendation that 'Scientific conclusions and business or policy decisions should not be based only on whether a p-value passes a specific threshold.' The authors justify their approach with respect to the analysis plan, but it would be preferable to follow the ASA recommendation. Rather than saying there was 'no effect', there was 'insufficient evidence of effect', I expect the authors do not want to add to the catalogue of 'absence of evidence is not evidence of absence' errors.

Reviewer #3: I don't have any remaining issues with this manuscript and would proceed with accepting it.

[LINK]

---

## [Editor Report · Decision Letter 2]

31 Jan 2020

Dear Dr Willis, 

On behalf of my colleagues and the academic editor, Dr. Sanjay Basu, I am delighted to inform you that your manuscript entitled "An adaptable implementation package targeting evidence-based indicators in primary care: a pragmatic cluster-randomised evaluation" (PMEDICINE-D-19-02682R2) has been accepted for publication in PLOS Medicine. 

PRODUCTION PROCESS

PRESS

PROFILE INFORMATION

Thank you again for submitting the manuscript to PLOS Medicine. We look forward to publishing it. 

Best wishes, 

Caitlin Moyer, Ph.D.

Associate Editor 

PLOS Medicine

plosmedicine.org